# Analysis of the Use of Korean Medicine Treatments among Children and Adolescents in South Korea: Analysis of Nationally Representative Sample

**DOI:** 10.3390/healthcare12040467

**Published:** 2024-02-13

**Authors:** Chan-Young Kwon

**Affiliations:** Department of Oriental Neuropsychiatry, College of Korean Medicine, Dong-Eui University, 52-57, Yangjeong-ro, Busanjin-gu, Busan 47227, Republic of Korea; beanalogue@deu.ac.kr

**Keywords:** integrative medicine, Korean medicine, conventional medicine, pediatrics, South Korea

## Abstract

Korean medicine (KM) is pivotal within South Korea’s healthcare system. This study aimed to investigate the current use and determinants associated with KM among children and adolescents through an analysis of the 2019 Korea Health Panel Annual Data. Subjects were divided into two groups: the integrative medicine (IM) group, utilizing both KM and conventional medicine (CM) (n = 163), and the CM-only group (n = 1843) for the year 2019. Differences in various factors between the IM and CM groups were investigated using the chi-square test or *t*-tests. Moreover, binomial logistic regression was employed to ascertain factors influencing the choice of KM over exclusive CM utilization. The IM group had a higher mean age (*p* = 0.011) and annual household income (*p* < 0.001) compared to the CM group. The regression analysis indicated a significant association between the use of both KM and CM and being an adolescent (*p* = 0.011), residing in Seoul/Gyeonggi/Incheon (*p* = 0.017), living in Daejeon/Chungcheong/Sejong (*p* = 0.001), and belonging to the first income percentile (*p* = 0.002). Significant differences were observed in the KM usage patterns between the groups of children and adolescents. These insights could contribute to the development of strategies for the optimal allocation of medical resources within South Korea’s distinctive medical framework.

## 1. Introduction

South Korea, alongside China, Taiwan, and Japan, officially incorporates East Asian traditional medicine (EATM) within its national healthcare system [1]. Korean medicine (KM), a form of EATM, is practiced by licensed doctors in the country, offering an approach distinct from conventional medicine (CM) [1,2]. Therefore, Koreans have the option to select KM, CM, or a combination of both, known as integrative medicine (IM), to enhance health and address diseases [2].

EATM is founded on a unique theoretical framework, promoting a holistic perspective of health [3]. Acupuncture and herbal medicine are primary treatments within EATM [3]. Engagement with KM services among children and adolescents in South Korea is notable. A 2017 national survey reported a 24.0% usage rate of KM services by this demographic in the preceding year (i.e., 2016) [4]. Moreover, the application of IM in these age groups is now recognized within the realm of evidence-based medicine [5].

The investigation of factors influencing medical service utilization among specific demographics is vital for the optimal deployment of healthcare resources. Studies on healthcare engagement during childhood are crucial for supporting a smooth healthcare transition to adulthood [6]. Previous research has indicated an association between KM utilization by parents and attitudes to and frequency of KM use by their children [4,7]. However, the determinants of KM service use by children and adolescents remain underexplored in South Korea. This study sought to fill this gap by evaluating the present use and associated factors influencing KM service utilization among children and adolescents through a comprehensive analysis of a nationally representative survey.

## 2. Materials and Methods

### 2.1. Data Source

The dataset for this investigation was obtained from the 2019 Korea Health Panel Annual Data (KHPAD-2019), a survey reflecting a national cross-section, administered by the Korea Institute for Health and Social Affairs in collaboration with the National Health Insurance Corporation. The composition of respondents included households and their members, selected through two-stage clustered probability sampling from a pool established by Statistics Korea. The survey was conducted on households and their members living in 17 cities and provinces in South Korea and included approximately 8500 households and their members. The survey obtained detailed information on healthcare utilization by dividing it into CM service, KM service, dental service, and health screening to provide a multifaceted view of healthcare utilization and expenses at the individual or household level. The determinants of healthcare utilization were divided into three categories, including socioeconomic factors, health state, and health behavior. Therefore, KHPAD-2019 can be considered an appropriate dataset to analyze healthcare utilization and related factors among households and household members in South Korea [8].

### 2.2. Subjects

The study population comprised individuals aged below 19 years. A total of 2346 children and adolescents were part of KHPAD-2019. Within this demographic, 2006 accessed outpatient services under CM in 2019. Of these, 1843 availed themselves solely of CM outpatient services, constituting the CM group. Meanwhile, 163 participants engaged with both CM and KM outpatient services, forming the IM group. Three subjects were excluded from the analysis because they exclusively used KM without CM (Figure 1).

### 2.3. Data Analysis

Analysis variables were selected based on the Andersen healthcare utilization model [9]. This theoretical model has contributed to core constructs for explaining health service utilization and is one of the most widely used models for analyzing factors related to the health care utilization by individuals [10]. This model has also been successfully applied in recent studies to analyze healthcare utilization in the pediatric population [11].

Predisposing factors included age, sex, and region, while enabling factors comprised household income, health insurance type, and actual loss insurance; the need factor was the presence of disability. Age categories were infants (0–1 years), children (2–11 years), and adolescents (12–18 years). This age classification was based on criteria from recent studies [12,13]. Sex was categorized as male or female. Regions were divided into Seoul/Gyeonggi/Incheon, Gangwon, Daejeon/Chungcheong/Sejong, Gwangju/Jeolla/Jeju, and Busan/Daegu/Ulsan/Gyeongsang. Annual household income was categorized into quartiles based on the criteria from a previous study [14]: first quartile (≥60 million won), second quartile (≥35.91 million won), third quartile (≥17.88 million won), and fourth quartile (<17.88 million won). Health insurance was categorized as either a subscription or medical aid, actual loss insurance as active or non-active, and disability as present or absent. Medical care for the population of interest in this study, particularly infants and children, is typically decided on by their parents. Therefore, it is important to examine parental factors such as annual household income and health insurance, which are parental socioeconomic factors. For example, in the case of health insurance, children and adolescents receive health insurance benefits as their parents’ dependents (i.e., employee insured or self-employed insured). Similarly, if the parents are eligible for medical aid, their children are also eligible for medical aid. Differences between the IM and CM groups were analyzed using chi-square analysis or *t*-tests. Furthermore, binomial logistic regression analysis was applied to identify factors associated with IM usage compared with exclusive CM usage. The outcomes are presented as odds ratios with 95% confidence intervals.

The analysis also investigated specific treatment details for the year 2019, including primary diagnoses for receiving KM or CM and the methods of KM treatment. The chi-square test was used to compare variable differences by age and group. This study, focusing solely on children and adolescents, did not apply the household member weights from the KHPAD-2019, reflecting the entire Korean populace. Statistical analyses were conducted using PASW Statistics for Windows, version 18.0 (SPSS Inc., Chicago, IL, USA), considering a *p*-value below 0.05 to denote statistical significance.

### 2.4. Ethical Considerations

The study was conducted in accordance with the Declaration of Helsinki, and it was approved by the Institutional Review Board of Dong-eui University Korean Medicine Hospital (DH-2023-08, approved on 13 November 2023).

## 3. Results

### 3.1. General Characteristics

The cohort for this investigation comprised the IM group (8.13%) and the CM group (91.87%). Age analysis revealed the mean age within the IM group to be statistically significantly greater than that of the CM group (*p* = 0.011). When broken down into age brackets, adolescents were significantly more prevalent in the IM group (*p* = 0.014), whereas infants were significantly less represented (*p* = 0.015). There was no significant intergroup difference in sex distribution.

Geographically, a higher proportion of the IM group resided in Daejeon/Chungcheong/Sejong (*p* = 0.004). Conversely, the CM group exhibited a greater presence in Gwangju/Jeolla/Jeju (*p* = 0.021) and Busan/Daegu/Ulsan/Gyeongsang (*p* = 0.006). The mean annual household income for the IM group was statistically significantly higher than that of the CM group (*p* < 0.001). This trend persisted when analyzing income percentiles, with the IM group showing a significantly higher representation in the top percentile (*p* < 0.001), while the remaining percentiles had a stronger presence in the CM group (*p* ranging from 0.002 to 0.016).

The analysis did not reveal any significant differences between the two groups concerning the type of health insurance, actual loss insurance, or the presence of disability. Financially, the IM group incurred higher overall medical expenses, including costs for CM treatments (*p* = 0.002), when compared with the CM group (Table 1).

### 3.2. Factors Associated with IM Service Use

Regression analysis was conducted to identify factors that were significantly associated with membership in the IM group as opposed to the CM group. The analysis determined that being an adolescent (*p* = 0.011), residing in Seoul/Gyeonggi/Incheon (*p* = 0.017), living in Daejeon/Chungcheong/Sejong (*p* = 0.001), and belonging to the first income percentile (*p* = 0.002) were significantly associated with inclusion in the IM group (Table 2).

### 3.3. Main Diagnosis in the IM Group

Within the IM group, the predominant diagnostic category for which KM treatment was administered was “other disorders”, accounting for 77.7% of cases; these disorders were not specifically delineated in the dataset. Analysis by age group revealed that musculoskeletal disorders were reported in approximately 50% of adolescents, a rate which was significantly higher than that of the children’s group (*p* < 0.001) (Table 3).

The primary diagnoses for which CM treatments were administered were compared between the IM and CM groups. For both groups, respiratory diseases emerged as the leading reason for CM intervention. The IM group had a significantly smaller proportion of respiratory disease cases, including the common cold (*p* < 0.001), relative to the CM group. Conversely, the IM group had higher proportions of patients treated for most other conditions (Table 4).

### 3.4. Details of KM Treatments in the IM Group

Analysis of KM treatments within the IM group indicated a predominant use of non-pharmacological methods, accounting for 75.1% of the treatments. Age-specific analysis showed that non-pharmacological treatments were significantly more prevalent among adolescents (*p* < 0.001) relative to children, whereas herbal medicine was used less frequently in the adolescent group (*p* < 0.001). Across all age groups, acupuncture was the most commonly employed individual KM treatment (58.9%), with herbal medicines being the second most utilized (29.8%) (Table 5).

## 4. Discussion

### 4.1. Interpretation of the Study Findings

This study aimed to investigate the current use and influencing factors of KM services among children and adolescents in South Korea. The data revealed that the proportion of the under-19 demographic using exclusively KM services was minimal (0.15%, 3/2009). Among the rest, 8.13% (163/2006) were utilizing both KM and CM. This low rate of KM usage in the under-19 age group corroborates previous studies that have indicated a lower frequency of KM service utilization among younger cohorts. The 2021 national survey on KM and herbal medicine usage found an average annual usage rate of 71.0% across all age groups in South Korea, with marked differences across ages: only 33.1% among those aged 19–29 years, compared with 90.6% among individuals over 60 years of age [15]. Consistent with this, the current study found that KM service use increased with age: 2.90% in infants, 7.60% in children, and 10.60% in adolescents.

A statistically significant difference in annual household income between the IM and CM groups was observed, with the IM group generally exhibiting higher incomes. Both the overall medical costs and the CM treatment costs were significantly higher for the IM group compared with the CM group. The regression analysis further highlighted economic status as a key determinant for inclusion in the IM group; specifically, individuals in the top income quartile were four times more likely to use IM services compared with those in the lowest quartile. Prior analyses of KHPAD-2019 also found a higher total income among KM users compared with CM users [14,16]. These observations contribute to the discourse on health disparities influenced by economic factors [17,18,19]. National policies aiming to mitigate such health inequalities could be informed by clinical practice guidelines for KM, which are being developed in line with the principles of evidence-based medicine. As of January 2024, clinical practice guidelines for KM applicable to pediatric conditions include those for growth disorders [20], autism spectrum disorders [21], and anorexia (in development).

Age-related differences in the use of KM services were noted. Specifically, among children, musculoskeletal diseases represented only 3.6% of the main KM treatment diagnoses, whereas the rate was 50.82% in the adolescent group. KHPAD-2019 does not detail conditions categorized as “other disorders.” However, when considering the primary diagnoses for CM treatments in the groups, it is anticipated that respiratory conditions would constitute a significant proportion of these “other disorders”, although existing research to confirm this assumption is lacking. Both groups predominantly sought CM treatment for respiratory diseases, with common colds being the most frequent diagnosis. Several potential reasons might explain why common colds were significantly less common as the primary diagnosis for CM treatment in the IM group compared with the CM group. First, KM, as part of EATM, often focuses on prevention, and the recent literature suggests that EATM treatments may reduce the incidence of respiratory ailments such as colds and influenza [22,23]. However, this dataset did not verify that the incidence among KM users was lower. Second, it is possible that individuals in the IM group may have opted solely for KM rather than CM services for treating common colds. A study analyzing national claims data indicated an increased use of KM services for common colds among Korean adults [24], but this did not account for CM use and thus did not fully explain our findings. Third, the possibility that the observed significant differences might be due to chance or unexplored variables within this dataset cannot be ruled out.

This study also highlighted that the types of KM treatments employed varied by age. Among children, the rates of non-pharmacological and pharmacological treatments were comparable (61.5% and 53.3%, respectively), whereas in the adolescent group, non-pharmacological treatments were more common (92.9% compared with 35.0% for pharmacological treatments). This trend appears to correlate with the predominance of musculoskeletal conditions in the adolescent group, aligning with the recognized efficacy of acupuncture for such conditions [25]. The factors behind the changes in KM treatment preferences among Korean adolescents warrant further investigation.

### 4.2. Differences from Existing Similar Studies

Previous attempts have been made to investigate factors related to the use of KM services in the children and adolescent population. For example, Kim et al. used the 2017 National Survey for the Usage of Korean Medicine as source data and compared 209 children who used KM services with 663 children who did not [4]. This study found that the parents’ knowledge, experience, and attitudes toward KM were significantly related to their children’s use of KM services [4]. However, the study had a limitation in that the children’s health factors were not reflected in the analysis because the children’s main disease leading to the use of KM services was not investigated [4]. On the other hand, our study attempted to interpret the health status of children and adolescents in relation to their healthcare utilization by analyzing the main diseases of the IM and CM groups. Importantly, our study analyzed CM use by the IM group and found that, as South Korea has a dual healthcare system with KM and CM, the pattern of KM use in this population may be dependent on the use of CM.

Lee et al., who analyzed the same dataset as the above study, also examined the effect of the parents’ experiences on their children’s KM use [7]. This study found that the parents’ experience with KM was associated with a 20% increase in their children’s KM use [7]. However, this study was also limited in that the children’s health factors were not considered for KM use [7]. Moreover, the children in this study were defined as those under 19 years of age, and no attempt was made to stratify them by age [7]. In contrast, our study found differences in the use of KM services between different age groups following age-based categorization of the children and adolescent population.

Countries other than South Korea have also investigated IM use among children and adolescents [26,27]. However, in South Korea, KM is part of the country’s main medical system, and some KM treatments (i.e., acupuncture, cupping, moxibustion, and Chuna therapy) are covered by national health insurance. Other KM treatments that are not covered by national health insurance may be covered by some private insurance providers. KM is a type of EATM; thus, populations in Asian countries are likely to be more familiar and favorable toward this type of treatment. However, even among Asian countries that practice EATM, there are differences in the medical systems including health insurance and licensing systems [1,2]. The revised Act on the Promotion of Korean Medicine and Pharmaceuticals, which came into effect in January 2024, specifies that the Korean government legally supports the development and use of KM [28]. Therefore, our study may be valuable as the findings could reflect differences in medical systems and cultural factors.

### 4.3. Limitations

This study was pioneering in its examination of the status and factors associated with KM service use among Korean children and adolescents. However, we recognize certain limitations. Being cross-sectional, the study did not consider changes in healthcare utilization over time. The design also restricts causal inferences between investigated variables and healthcare utilization [29]. Moreover, the limited scope of the study’s data prevented a more detailed analysis. For example, the main diagnoses leading to KM treatment and other potentially significant variables were not specifically investigated in KHPAD-2019. Furthermore, important need factors, such as subjective health status [30], stress perception, depression, anxiety [31,32], and chronic disease presence [33], while assessed for adults in KHPAD-2019, were not assessed for children and adolescents; therefore, they were not factored into this study’s analysis. The disproportionate distribution of children according to age is also a limitation of this study. In particular, the included sample size of infants was small, which led to low resolution of statistically significant results.

## 5. Conclusions

Our research indicates that, in South Korea, 8.13% of the population under 19 uses IM, combining both CM and KM services, with the likelihood of use increasing with age. A significant factor influencing the utilization of KM services was found to be economic status, with those in higher economic brackets more likely to engage with KM. This association between KM use and economic capacity may point to potential health disparities in medical care access among children and adolescents. Our study also found age-related differences in the use of KM services, with KM use due to musculoskeletal disorders appearing to increase with age in this population. Additionally, with increasing age, the rate of non-pharmacological KM treatment was increased. There were also differences in the use of CM services between the IM and CM groups, with the most notable difference being the use of CM for treating respiratory diseases was significantly lower in the IM group. These findings may contribute to a better understanding of the factors associated with KM use among children and adolescents in South Korea.

## Figures and Tables

**Figure 1 healthcare-12-00467-f001:**
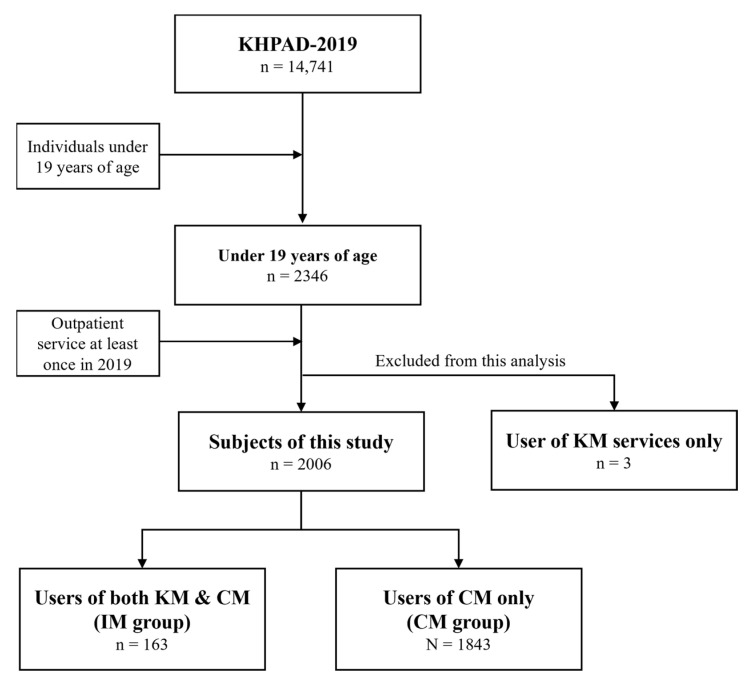
Study sample inclusion flow chart. Abbreviations. CM, conventional medicine; IM, integrative medicine; KHPAD-2019, the 2019 Korea Health Panel Annual Data; KM, Korean medicine.

**Table 1 healthcare-12-00467-t001:** Differences in basic demographic variables between the IM group and the CM group.

Group	Variables	IM (n = 163)	CM (n = 1843)	*p*	
Age	0–1 (infant)	4 (2.5%)	134 (7.3%)	0.015 *	0.006 **
	2–11 (children)	99 (60.7%)	1203 (65.3%)	0.266	
	12–18 (adolescent)	60 (36.8%)	506 (27.5%)	0.014 *	
	Mean age (year)	9.40 ± 4.927	8.36 ± 4.894	0.011 *	
Sex	Men	90 (55.2%)	949 (51.5%)	0.370	
	Women	73 (44.8%)	894 (48.5%)		
Region	Seoul/Gyeonggi/Incheon	60 (36.8%)	547 (29.7%)	0.062	0.001 **
	Gangwon	5 (3.1%)	49 (2.7%)	0.798	
	Daejeon/Chungcheong/Sejong	48 (29.4%)	362 (19.6%)	0.004 **	
	Gwangju/Jeolla/Jeju	23 (14.1%)	401 (21.8%)	0.021 *	
	Busan/Daegu/Ulsan/Gyeongsang	27 (16.6%)	484 (26.3%)	0.006 **	
House income	1st percentile	111 (68.1%)	784 (42.5%)	<0.001 ***	<0.001 ***
	2nd percentile	36 (22.1%)	623 (33.8%)	0.002 **	
	3rd percentile	10 (6.1%)	231 (12.5%)	0.016 *	
	4th percentile	6 (3.7%)	205 (11.1%)	0.002 **	
	Mean income (KRW/year)	7361.90 ± 3223.32	5844.17 ± 3874.77	<0.001 ***	
Health insurance type	Employee or local	161 (98.8%)	1779 (96.5%)	0.166	
	Medical aid or others	2 (1.2%)	64 (3.5%)		
Actual loss insurance	Active	123 (75.5%)	1289 (69.9%)	0.152	
	Non-active	40 (24.5%)	554 (30.1%)		
Disability	Yes	1 (0.6%)	23 (1.2%)	0.716	
	No	162 (99.4%)	1820 (98.8%)		
Annual medical expenses	KM treatments (KRW/year)	272,172.58 ± 525,459.86	na	na	
	CM treatments (KRW/year)	254,465.98 ± 333,949.48	159,596.66 ± 372,210.23	0.002 **	

*, *p* < 0.05; **, *p* < 0.01; ***, *p* < 0.001. Abbreviations: CM, conventional medicine; IM, integrative medicine; KM, Korean medicine; na, not applicable.

**Table 2 healthcare-12-00467-t002:** Factors associated with the use of IM in children and adolescents.

Group	Variables	OR (95% LLCI to ULCI)	*p*
Age (ref: 0–1)	2–11	2.61 (0.94 to 7.26)	0.066
	12–18	3.87 (1.37 to 10.92)	0.011 *
Sex (ref: women)	men	1.10 (0.79 to 1.53)	0.563
Region (ref: Busan/Daegu/Ulsan/Gyeongsang)	Seoul/Gyeonggi/Incheon	1.79 (1.11 to 2.89)	0.017 *
	Gangwon	1.86 (0.67 to 5.15)	0.235
	Daejeon/Chungcheong/Sejong	2.30 (1.40 to 3.79)	0.001 **
	Gwangju/Jeolla/Jeju	1.02 (0.57 to 1.82)	0.945
Income (ref: 4th percentile)	1st percentile	4.07 (1.71 to 9.70)	0.002 **
	2nd percentile	1.79 (0.72 to 4.42)	0.208
	3rd percentile	1.27 (0.44 to 3.60)	0.659
Health insurance type (ref: medical aid or others)	Employee or local	1.47 (0.33 to 6.50)	0.610
Actual loss insurance (ref: non-active)	Active	1.26 (0.86 to 1.84)	0.236
Disability (ref: no)	Yes	0.60 (0.08 to 4.61)	0.621

*, *p* < 0.05; **, *p* < 0.01. Abbreviations: IM, integrative medicine; LLCI, lower limit of the confidence interval; OR, odds ratio; ULCI, upper limit of the confidence interval.

**Table 3 healthcare-12-00467-t003:** Main diagnoses for using KM treatments in the IM group.

Main Diagnosis		Valid Cases (n = 462)	Age Groups			*p* ^a^
			Infant (n = 4)	Children (n = 275)	Adolescent (n = 183)	
Musculoskeletal disorders	Total	103 (22.3%)	0 (0%)	10 (3.6%)	93 (50.8%)	<0.001 ***
	Back pain	5 (1.1%)	0 (0%)	0 (0%)	5 (2.7%)	0.010 *
	IVDD	18 (3.9%)	0 (0%)	0 (0%)	18 (9.8%)	<0.001 ***
	Other joint disorders	79 (17.1%)	0 (0%)	10 (3.6%)	69 (37.7%)	<0.001 ***
	Arthritis	1 (0.2%)	0 (0%)	0 (0%)	1 (0.6%)	0.400
Others	Total	359 (77.7%)	4 (100%)	265 (96.4%)	90 (49.2%)	<0.001 ***

*, *p* < 0.05; ***, *p* < 0.001; *p*-value ^a^, comparison between children and adolescent groups. Abbreviations: IM, integrative medicine; IVDD, intervertebral disc disease; KM, Korean medicine; na, not applicable.

**Table 4 healthcare-12-00467-t004:** Main diagnoses for using CM treatments in the two groups.

Main Diagnosis		Valid IM Cases (n = 3165)	Valid CM Cases (n = 25,642)	*p*
Respiratory diseases	Total	2042 (64.5%)	18,912 (73.8%)	<0.001 ***
	Asthma	7 (0.2%)	63 (0.3%)	>0.999
	Pulmonary emphysema	0 (0%)	1 (0.0%)	>0.999
	Bronchiectasis	0 (0%)	1 (0.0%)	>0.999
	Pneumonia/bronchitis	51 (1.6%)	552 (2.2%)	0.051
	Common cold	1984 (62.7%)	18,295 (71.4%)	<0.001 ***
Eye disease	Total	105 (3.3%)	843 (3.3%)	0.921
Digestive system disease	Total	108 (3.4%)	631 (2.5%)	0.002 **
	Hepatitis	0 (0%)	1 (0.0%)	>0.999
	Enteritis	85 (2.7%)	574 (2.2%)	0.113
	Gastritis	12 (0.4%)	56 (0.2%)	0.082
	Hemorrhoids	11 (0.4%)	0 (0%)	<0.001 ***
Musculoskeletal disorders	Total	111 (3.5%)	368 (1.4%)	<0.001 ***
	Knee arthrosis	23 (0.7%)	2 (0.0%)	<0.001 ***
	Shoulder joint disorder	1 (0.0%)	4 (0.0%)	0.441
	IVDD	14 (0.4%)	17 (0.1%)	<0.001 ***
	Back pain	0 (0%)	2 (0.0%)	>0.999
	Other spinal disease	1 (0.0%)	20 (0.1%)	0.722
	Other joint pain and muscle pain	72 (2.3%)	323 (1.3%)	<0.001 ***
Other	Total	799 (25.2%)	4888 (19.1%)	<0.001 ***
	Hypothyroidism	0 (0%)	14 (0.1%)	0.389
	Hyperthyroidism	0 (0%)	10 (0.0%)	0.614
	Stroke	0 (0%)	181 (0.7%)	<0.001 ***
	Cancer	0 (0%)	70 (0.3%)	<0.001 ***
	Gynecological diseases	0 (0%)	26 (0.1%)	0.107
	Unipolar/bipolar disorder	9 (0.3%)	34 (0.1%)	0.048 *
	Urinary disorders	3 (0.1)	55 (0.2%)	0.206
	Headache	10 (0.3%)	29 (0.1%)	0.008 **
	Unspecified fever	13 (0.4%)	117 (0.5%)	0.888
	Dizziness	6 (0.2%)	15 (0.1%)	0.022 *
	Other disorders	758 (24.0%)	4337 (16.9%)	<0.001 ***

*, *p* < 0.05; **, *p* < 0.01; ***, *p* < 0.001. Abbreviations: CM, conventional medicine; IM, integrative medicine; IVDD, intervertebral disc disease.

**Table 5 healthcare-12-00467-t005:** Use of KM treatments in children and adolescents.

KM Treatments		Valid Cases (n = 679)	Age Groups			*p* ^a^
			Infant (n = 6)	Children (n = 379)	Adolescent (n = 294)	
Non-pharmacological treatments	Total	510 (75.1%)	4 (66.7%)	233 (61.5%)	273 (92.9%)	<0.001 ***
	Acupuncture	400 (58.9%)	1 (16.7%)	172 (45.4%)	227 (77.2%)	<0.001 ***
	Moxibustion	44 (6.5%)	0 (0%)	25 (6.6%)	19 (6.5%)	>0.999
	Cupping	27 (4.0%)	1 (16.7%)	6 (1.6%)	20 (6.8%)	0.001 **
	PA	41 (6.0%)	0 (1%)	10 (2.6%)	31 (10.5%)	<0.001 ***
	Chuna	26 (3.8%)	0 (0%)	0 (0%)	26 (8.8%)	<0.001 ***
	Manual therapy	11 (1.6%)	0 (0%)	0 (0%)	11 (3.7%)	<0.001 ***
	Other PTs	176 (25.9%)	4 (66.7%)	65 (17.2%)	107 (36.4%)	<0.001 ***
Oral herbal medicine	Total	309 (45.5%)	4 (66.7%)	202 (53.3%)	103 (35.0%)	<0.001 ***
	HD	202 (29.8%)	1 (16.7%)	142 (37.5%)	59 (20.1%)	<0.001 ***
	Expensive HPs ^b^	3 (0.4%)	1 (16.7%)	2 (0.5%)	0 (0%)	0.507
	General HPs	121 (17.8%)	2 (33.3%)	69 (18.2%)	50 (17.0%)	0.760

**, *p* < 0.01; ***, *p* < 0.001; *p*-value ^a^, comparison between children and adolescent groups; Expensive HPs ^b^, expensive herbal preparations (e.g., Gongjindan) for use as a health tonic rather than for disease treatment. The medical panel survey differentiated between expensive and general herbal preparations for disease treatment. Abbreviations: HD, herbal decoction; HP, herbal preparation; KM, Korean medicine; PA, pharmacopuncture; PT, physical therapy.

## Data Availability

The data presented in this study are available on request from the corresponding author.

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
