# Peer review of "Analysis of the Use of Korean Medicine Treatments among Children and Adolescents in South Korea: Analysis of Nationally Representative Sample"

_healthcare, 2024, doi:10.3390/healthcare12040467_

Round 1
Reviewer 1 Report
Comments and Suggestions for Authors
The authors of this study have conducted an analysis on the utilization of Korean Medicine Treatments among children and adolescents in South Korea. This title is worthy of investigation. However, regarding reviewing the manuscript some comments are mentioned below that need more explanation, clarification, or modification.
Keywords:
1. The keywords should be selected wisely and carefully. East Asian traditional medicine and the 2019 Korea Health Panel Annual Data are too long and not suitable to be keywords.
Introduction:
1. Regarding lines 33 and 34, mentioning the meridian theory, as well as yin-yang and the five elements theory without declaring a brief definition of these theories is not appropriate. Please reconsider it.
2. Regarding line 38 and the phrase “… within the preceding year.”, please clarify exactly the years that were investigated.
Methods:
1. Regarding lines 72 and 73, on what basis did you categorize the age of patients? Can you provide an evidence-based article for that?
2. Infants does not have any power to select their remedies and their parents define it for them. So how can you evaluate and analyze their data? Please explain that.
Results:
1. Regarding to table1, it is better that every group has a single p-value, along with the categorized p-value, that compares the variable generally with type of medicine. For instance the relevance of type of medicine with religion, age, house income, etc. as a whole could be mentioned.
2. The structure of table 3, specifically the others category is not fine. Why some diseases, i.e. stroke, facial nerve disorder, etc., that was not present in any cases (N=0) was placed in the table? The number of other disorders and total are the same. Table needs to be more informative and you should not just try to fill out the table.
3. According to table 4, how did you categorize unspecified fever as a respiratory disease?
4. What do you mean by general and expensive herbal preparation? Please clarify it.
Discussion:
1. All the information and results that are provided in this section should be discussed and compared with other similar or relatively similar studies. Is some parts, only the pure results were presented without any comparison with other surveys; so it should be modified.
2. Please provide more references to present the concept of your study better. References are not sufficient.
A minor revision regarding the English language grammar and vocabulary proficiency seems to be essential.
Wish you luck and prosperity in preparing the manuscript.

Comments on the Quality of English Language
A minor revision regarding the English language grammar and vocabulary proficiency seems to be essential.
Author Response
- Response to Comments from Reviewer 1
Overall comment:
The authors of this study have conducted an analysis on the utilization of Korean Medicine Treatments among children and adolescents in South Korea. This title is worthy of investigation. However, regarding reviewing the manuscript some comments are mentioned below that need more explanation, clarification, or modification.
Response:
Thank you very much for taking your valuable time to review this manuscript. We have no doubt that the reviewer’s comments will help to further improve the quality of this manuscript.
Keywords_Comment 1:
The keywords should be selected wisely and carefully. East Asian traditional medicine and the 2019 Korea Health Panel Annual Data are too long and not suitable to be keywords.
Response:
Thank you for your comment. The keywords were changed as follows.
“Keywords: Integrative medicine; Korean medicine; conventional medicine; Pediatrics; South Korea.”
(Please refer Page 1, red words)
Introduction_Comment 1:
Regarding lines 33 and 34, mentioning the meridian theory, as well as yin-yang and the five elements theory without declaring a brief definition of these theories is not appropriate. Please reconsider it.
Response:
Thank you for your comment and I agree with. I think that the meridian theory and the five elements theory do not need to be dealt with in depth in this study, which is likely to cause confusion to readers. Therefore, I have removed the description of that theory from this revised manuscript.
“EATM is founded on a unique theoretical framework, promoting a holistic perspective of health [3].”
(Please refer Page 1)
Introduction_Comment 2:
Regarding line 38 and the phrase “… within the preceding year.”, please clarify exactly the years that were investigated.
Response:
Thank you for your comment. I modified the sentence to make its meaning clear as follows.
“A 2017 national survey reported a 24.0% usage rate of KM services by this demographic within the preceding year (i.e., 2016) [4].”
(Please refer Page 1, red words)
Methods_Comment 1:
Regarding lines 72 and 73, on what basis did you categorize the age of patients? Can you provide an evidence-based article for that?
Response:
Thank you for your comment. I have added citations to two recent studies using the same classification in this revised manuscript.
“This age classification was based on criteria from recent studies [12,13].”
(Please refer Page 3, red words)
Methods_Comment 2:
Infants does not have any power to select their remedies and their parents define it for them. So how can you evaluate and analyze their data? Please explain that.
Response:
Thank you for your comment. I have added statements explaining the importance of parental factors in health care use in this population.
“The medical care of the population of interest in this study, particularly infants and children, is typically decided by their parents. Therefore, it is important to examine parental factors such as annual household income and health insurance, which are parental socioeconomic factors. For example, in the case of health insurance, children and adolescents receive health insurance benefits as dependents from their parents (i.e., employee insured or self-employed insured). Similarly, if the parents are eligible for medical aid, their children are also eligible for medical aid.”
(Please refer Page 3, red words)
Results_Comment 1:
Regarding to table1, it is better that every group has a single p-value, along with the categorized p-value, that compares the variable generally with type of medicine. For instance the relevance of type of medicine with religion, age, house income, etc. as a whole could be mentioned.
Response:
Thank you for your comment. As the reviewer commented, I added p-values.
(Please refer Page 4, Table 1, red words)
Results_Comment 2:
The structure of table 3, specifically the others category is not fine. Why some diseases, i.e. stroke, facial nerve disorder, etc., that was not present in any cases (N=0) was placed in the table? The number of other disorders and total are the same. Table needs to be more informative and you should not just try to fill out the table.
Response:
Thank you for your comment. I deleted the unnecessary information pointed out by the reviewer from Table 3.
(Please refer Page 5, Table 3)
Results_Comment 3:
According to table 4, how did you categorize unspecified fever as a respiratory disease?
Response:
Thank you for your comment. As the reviewer pointed out, it is appropriate that unspecified fever should be classified in the category ‘Other’. Therefore, in this revised manuscript, this has been corrected and the related statistics have also been changed.
(Please refer Page 6, Table 4, red words)
Results_Comment 4:
What do you mean by general and expensive herbal preparation? Please clarify it.
Response:
Thank you for your comment. In South Korea, some expensive herbal preparations such as Gongjindan are sometimes used for nourishment, rather than for therapeutic purposes. In this panel dataset, expensive HP for nourishment and general HP for disease treatment were separately investigated. I have added this to this revised manuscript.
“Expensive HPsb, expensive herbal preparations (e.g., Gongjindan) for use as a health tonic rather than for disease treatment. The medical panel survey differentiated between expensive and general herbal preparations for disease treatment.”
(Please refer Page 7, red words)
Discussion_Comment 1:
All the information and results that are provided in this section should be discussed and compared with other similar or relatively similar studies. Is some parts, only the pure results were presented without any comparison with other surveys; so it should be modified.
Response:
Thank you for your comment. As pointed out by the reviewer, I created a separate subheading in 4. Discussion and discussed the meaning of the differences between existing studies and current findings.
“4.2. Differences from Existing Similar Studies
Previous attempts have been made to investigate factors related to the use of KM services in the children and adolescent population. For example, Kim et al. used the 2017 National Survey for the Usage of Korean Medicine as source data and compared 209 children who used KM services with 663 children who did not [4]. This study found that the parents' knowledge, experience, and attitudes toward KM were significantly related to their children's use of KM services [4]. However, the study had a limitation in that the children's health factors were not reflected in the analysis because the children's main disease leading to the use of KM services was not investigated [4]. On the other hand, our study attempted to interpret the health status of children and adolescents in relation to their healthcare utilization by analyzing the main diseases of the IM and CM groups. Importantly, our study analyzed CM use by the IM group and found that, as South Korea has a dual healthcare system with KM and CM, the pattern of KM use in this population may be dependent on the use of CM.
Lee et al., who analyzed the same dataset as the above study, also examined the effect of the parents' experiences on their children's KM use [7]. This study found that the parents' experience with KM was associated with a 20% increase in their children's KM use [7]. However, this study was also limited in that the children's health factors were not considered for KM use [7]. Moreover, the children in this study were defined as those under 19 years of age, and no attempt was made to stratify them by age [7]. In contrast, our study found differences in the use of KM services between different age groups following age-based categorization of the children and adolescent population.
Countries other than South Korea have also investigated IM use among children and adolescents [26,27]. However, in South Korea, KM is part of the country's main medical system, and some KM treatments (i.e., acupuncture, cupping, moxibustion, and Chuna therapy) are covered by the national health insurance. Other KM treatments that are not covered by the national health insurance may be covered by some private insurance providers. KM is a type of EATM; thus, populations in Asian countries are likely to be more familiar and favorable toward this type of treatment. However, even among Asian countries that practice EATM, there are differences in the medical systems including health insurance and licensing systems [1,2]. The revised Act on the Promotion of Korean Medicine and Pharmaceuticals, which came into effect in January 2024, specifies that the Korean government legally supports the development and use of KM [28]. Therefore, our study may be valuable as the findings could reflect differences in medical systems and cultural factors.”
(Please refer Pages 8-9, red words)
Discussion_Comment 2:
Please provide more references to present the concept of your study better. References are not sufficient.
Response:
Thank you for your comment. Appropriate references have been added to the discussion. The added references are as follows [26-31]:
- Groenewald, C.B.; Beals-Erickson, S.E.; Ralston-Wilson, J.; Rabbitts, J.A.; Palermo, T.M. Complementary and Alternative Medicine Use by Children With Pain in the United States. Acad Pediatr 2017, 17, 785-793, doi:10.1016/j.acap.2017.02.008.
- Zuzak, T.J.; Boňková, J.; Careddu, D.; Garami, M.; Hadjipanayis, A.; Jazbec, J.; Merrick, J.; Miller, J.; Ozturk, C.; Persson, I.A.; et al. Use of complementary and alternative medicine by children in Europe: published data and expert perspectives. Complement Ther Med 2013, 21 Suppl 1, S34-47, doi:10.1016/j.ctim.2012.01.001.
- Jeong, H.I.; Kim, K.H.; Yi, J.; Kim, D.; Sung, S.-H.; Lee, E.-S. A Review and Implication of Act on the Promotion of Korean Medicine and Pharmaceuticals. Journal of Society of Preventive Korean Medicine 2022, 26, 69-74.
- Wang, X.; Cheng, Z. Cross-Sectional Studies: Strengths, Weaknesses, and Recommendations. Chest 2020, 158, S65-s71, doi:10.1016/j.chest.2020.03.012.
- Kwak, S.; Lee, Y.; Baek, S.; Shin, J. Effects of Subjective Health Perception on Health Behavior and Cardiovascular Disease Risk Factors in Patients with Prediabetes and Diabetes. Int J Environ Res Public Health 2022, 19, doi:10.3390/ijerph19137900.
- Li, H.; Pang, M.; Wang, J.; Xu, J.; Kong, F. Effects of Health Service Utilization and Informal Social Support on Depression, Anxiety, and Stress among the Internal Migrant Elderly following Children in Weifang, China. Int J Environ Res Public Health 2022, 19, doi:10.3390/ijerph192214640.
Other Comment:
A minor revision regarding the English language grammar and vocabulary proficiency seems to be essential.
Wish you luck and prosperity in preparing the manuscript.
Response:
Thank you for your comment. Before submitting this revised manuscript, we engaged the services of a professional English editing company to correct grammar and vocabulary. Related certification was also submitted.

Reviewer 2 Report
Comments and Suggestions for Authors
The paper is well written and is focused on the field which yet, has not been sufficiently studied. From my perspective I consider the paper almost ready to be published, provided the following 2 issues are improved:
1. Please better describe the methods employed, i.e. provide more details on data source (2.1) and explain why selected methods are suitable for this research, i.e. why these methods were used and not the others. The choice is correct, but please explain better why.
2. Please find on more conclusions from this study which should be provided in last section.
Author Response
- Response to Comments from Reviewer 2
Overall comment:
The paper is well written and is focused on the field which yet, has not been sufficiently studied. From my perspective I consider the paper almost ready to be published, provided the following 2 issues are improved:
Response:
Thank you very much for taking your valuable time to review this manuscript. We have no doubt that the reviewer’s comments will help to further improve the quality of this manuscript.
Comment 1:
Please better describe the methods employed, i.e. provide more details on data source (2.1) and explain why selected methods are suitable for this research, i.e. why these methods were used and not the others. The choice is correct, but please explain better why.
Response:
Thank you for your comment. I've added a description of this dataset. Additionally, I have added an explanation justifying the use of the Andersen healthcare utilization model, which was used as the analysis model for this study.
“The dataset for this investigation was obtained from the 2019 Korea Health Panel Annual Data (KHPAD-2019), a survey reflecting a national cross-section, administered by the Korea Institute for Health and Social Affairs in collaboration with the National Health Insurance Corporation. The composition of respondents included households and their members, selected through two-stage clustered probability sampling from a pool established by Statistics Korea. The survey was conducted on households and their members living in 17 cities and provinces in South Korea and included approximately 8,500 households and their members. The survey obtained detailed information on healthcare utilization by dividing it into CM service, KM service, dental service, and health screening to provide a multifaceted view of healthcare utilization and expenses at the individual or household level. The determinants of healthcare utilization were divided into three categories, including socioeconomic factors, health state, and health behavior. Therefore, KHPAD-2019 can be considered an appropriate dataset to analyze healthcare utilization and related factors among households and household members in South Korea [8].”
(Please refer Page 2, red words)
“Analysis variables were selected based on the Andersen healthcare utilization model [9]. This theoretical model has contributed to core constructs for explaining health service utilization and is one of the most widely used models for analyzing factors related to the health care utilization of individuals [10]. This model has also been successfully applied in recent studies to analyze healthcare utilization in the pediatric population [11].”
(Please refer Page 3, red words)
Comment 2:
Please find on more conclusions from this study which should be provided in last section.
Response:
Thank you for your comment. In this revised manuscript, we have added the following conclusions.
“Our research indicates that in South Korea, 8.13% of the population under 19 uses IM, combining both CM and KM services, with the likelihood of use increasing with age. A significant factor influencing the utilization of KM services was found to be economic status, with those in higher economic brackets more likely to engage with KM. This association between KM use and economic capacity may point to potential health disparities in medical care access among children and adolescents. Our study also found age-related differences in the use of KM services, with KM use due to musculoskeletal disorders appearing to increase with age in this population. Additionally, with increasing age, the rate of non-pharmacological KM treatment was increased. There were also differences in the use of CM services between the IM and CM groups, with the most notable difference being that the use of CM for treating respiratory diseases was significantly lower in the IM group. These findings may contribute to a better understanding of the factors associated with KM use among children and adolescents in South Korea.”
(Please refer Page 9, red words)

Reviewer 3 Report
Comments and Suggestions for Authors
Dear author(s),
Your manuscript is very interesting, accurate and comprehensive. Mostly, I do not have too many suggestions of improvement and they do not relate to the methodology employed, but rather to the introduction and discussion sections.
Kindly, see below what are my suggestions of improvement:
- In Table 1 you have inserted a few variables which do not necessarily apply to your target group, namely children and adolescents. For instance, health insurance type with the categories employee or local and medical aid or others, since you focus on children and adolescents they cannot be employees. If you still want to keep this variable, I would add parents’ health insurance type. Actual loss insurance is something I was wondering if it is relevant in this context. What I was wondering is the fact that in most countries all children and adolescents have public health insurance until they reach the age of 18. Is it the case of South Korea? In which way is different? Again, if you want to keep this variable I would add parents or explain a bit more how a child or an adolescent in South Korea can loose his health insurance.
- The following suggestions apply for the Introduction and the Discussion sections: Can the author(s) connect the usage of KM to the cultural background of the Korean population? Is there any difference between KM, and other complementary medicine in China, Taiwan and Japan, in these countries and territory? Are there any laws which support more KM and IM rather than CM (Conventional Medicine)? I will explain a bit why these questions crossed my mind and it is all connected to the culture and laws. For example, in Romania, laws are in favour of conventional medicine and we barely talk about acupuncture. If one person would like to choose Complementary Medicine, it will be very expensive, so basically there is no need to pay more if you can use drugs to get well.
- For the limitations paragraph I would also recommend to include another limitation, namely the disproportionate distribution of children by age. For instance, infants were not well represented, and it was common sense to have a significant result.
Thank you!
Good luck!
Author Response
- Response to Comments from Reviewer 3
Overall comment:
Your manuscript is very interesting, accurate and comprehensive. Mostly, I do not have too many suggestions of improvement and they do not relate to the methodology employed, but rather to the introduction and discussion sections.
Kindly, see below what are my suggestions of improvement:
Response:
Thank you very much for taking your valuable time to review this manuscript. We have no doubt that the reviewer’s comments will help to further improve the quality of this manuscript.
Comment 1:
In Table 1 you have inserted a few variables which do not necessarily apply to your target group, namely children and adolescents. For instance, health insurance type with the categories employee or local and medical aid or others, since you focus on children and adolescents they cannot be employees. If you still want to keep this variable, I would add parents’ health insurance type. Actual loss insurance is something I was wondering if it is relevant in this context. What I was wondering is the fact that in most countries all children and adolescents have public health insurance until they reach the age of 18. Is it the case of South Korea? In which way is different? Again, if you want to keep this variable I would add parents or explain a bit more how a child or an adolescent in South Korea can loose his health insurance.
Response:
Thank you for your comment. I have added explanations about the public health insurance and actual loss insurance commented as follows. It also adds to the importance of examining parental factors in examining health care utilization in this population.
“The medical care of the population of interest in this study, particularly infants and children, is typically decided by their parents. Therefore, it is important to examine parental factors such as annual household income and health insurance, which are parental socioeconomic factors. For example, in the case of health insurance, children and adolescents receive health insurance benefits as dependents from their parents (i.e., employee insured or self-employed insured). Similarly, if the parents are eligible for medical aid, their children are also eligible for medical aid.”
(Please refer Page 3, red words)
Comment 2:
The following suggestions apply for the Introduction and the Discussion sections: Can the author(s) connect the usage of KM to the cultural background of the Korean population? Is there any difference between KM, and other complementary medicine in China, Taiwan and Japan, in these countries and territory? Are there any laws which support more KM and IM rather than CM (Conventional Medicine)? I will explain a bit why these questions crossed my mind and it is all connected to the culture and laws. For example, in Romania, laws are in favour of conventional medicine and we barely talk about acupuncture. If one person would like to choose Complementary Medicine, it will be very expensive, so basically there is no need to pay more if you can use drugs to get well.
Response:
Thank you for your comment. Based on the comments, I added the cultural aspects, health insurance aspects, and legal aspects.
“Countries other than South Korea have also investigated IM use among children and adolescents [26,27]. However, in South Korea, KM is part of the country's main medical system, and some KM treatments (i.e., acupuncture, cupping, moxibustion, and Chuna therapy) are covered by the national health insurance. Other KM treatments that are not covered by the national health insurance may be covered by some private insurance providers. KM is a type of EATM; thus, populations in Asian countries are likely to be more familiar and favorable toward this type of treatment. However, even among Asian countries that practice EATM, there are differences in the medical systems including health insurance and licensing systems [1,2]. The revised Act on the Promotion of Korean Medicine and Pharmaceuticals, which came into effect in January 2024, specifies that the Korean government legally supports the development and use of KM [28]. Therefore, our study may be valuable as the findings could reflect differences in medical systems and cultural factors.”
(Please refer Pages 8-9, red words)
Comment 3:
For the limitations paragraph I would also recommend to include another limitation, namely the disproportionate distribution of children by age. For instance, infants were not well represented, and it was common sense to have a significant result.
Response:
Thank you for your comment. The following has been added to reflect the reviewer's comments:
“The disproportionate distribution of children according to age is also a limitation of this study. In particular, the included sample size of infants was small, which led to low resolution of statistically significant results.”
(Please refer Page 9, red words)

Round 2
Reviewer 1 Report
Comments and Suggestions for Authors
Thanks for all your efforts and addressing the comments and recommendations that were provided.